# Towards Subject-Consistent and Text-Aligned Personalized Image Generation via Precise Attribute Learning

## Abstract

Recent advances in personalized image generation using Diffusion Transformers (DiTs) have shown remarkable progress. However, existing approaches face a trade-off between textual alignment and maintaining reference subjects. This issue primarily stems from the fact that directly injecting subject tokens may disrupt the sampling trajectory of the base model, while the methods through textual inversion struggle to capture detailed attributes of the subject. To address these limitations, we introduce a DiT based subject-driven generation framework **Genova** with an innovative attribute learning module. This attribute learning module integrates subject image tokens to improve the text-stream modulation, enhancing the representation of the subject's visual attributes distinctly. Contrary to traditional modulation techniques in DiTs, our proposed framework leverages the hierarchical features from the subject image tokens, facilitating more effective attribute learning. This enhancement allows for precise semantic understanding of the subject, thereby optimizing the model's inherent capabilities for textual alignment and enabling more flexible and controllable image generation. Moreover, we develop a synthetic dataset **CoupleX** featuring subject-paired samples that focus on depicting the activities and interactions within natural scenes, providing a richer context than previous datasets. Extensive experiments demonstrate that our method outperforms current state-of-the-art methods and achieves subject and prompt consistent personalized image generation.

## 1 Introduction

Foundational text-to-image models and related technologies have achieved significant advancements (Ho et al., 2020; Rombach et al., 2022; Zhang et al., 2023; Labs, 2024; Peebles & Xie, 2023), enabling the creation of highly realistic and diverse images from text descriptions. In recent years, personalized image generation (Ye et al., 2023; Xiao et al., 2025; Chen et al., 2024; Ruiz et al., 2023b) that creates images conforming to both textual semantics and reference subjects has gained significant attention across academic and industrial communities. This task merges the flexibility of text control with the precision of visual controls, providing the foundational infrastructure for diverse real-world applications, including commercial poster design and photo editing.

Recently, personalized image generation methods can be broadly categorized into two types based on their methods: (a) specialized text learning and (b) subject token injection. However, both methods face a trade-off when addressing the dual challenges of textual alignment and subject consistency (shown in Fig. 1). The first-type methods (Ruiz et al., 2023a; Gal et al., 2022; Chen et al., 2023; Garibi et al., 2025; Chen et al., 2025a; Zhong et al., 2025) center on learning specialized text embeddings, primarily by mapping the subject concept into the text latent space. This method exploits the language model's ability to interpret the reference subject, leveraging the base model's inherent advantages in textual alignment. However, learning an effective text embedding becomes challenging when the personalized subject features complex textures, causing these methods to struggle with maintaining subject consistency. For example, in Fig. 1 (a), the model only learns the attribute of "white petals" but fails to capture the attributes of "sparse stamens" and "green leaves". The second-type methods (Chen et al., 2025b; Wu et al., 2025; Tan et al., 2025; Ye et al., 2023; Li et al., 2023; Mou et al., 2025) involve injecting subject tokens into pre-trained text-to-image models to

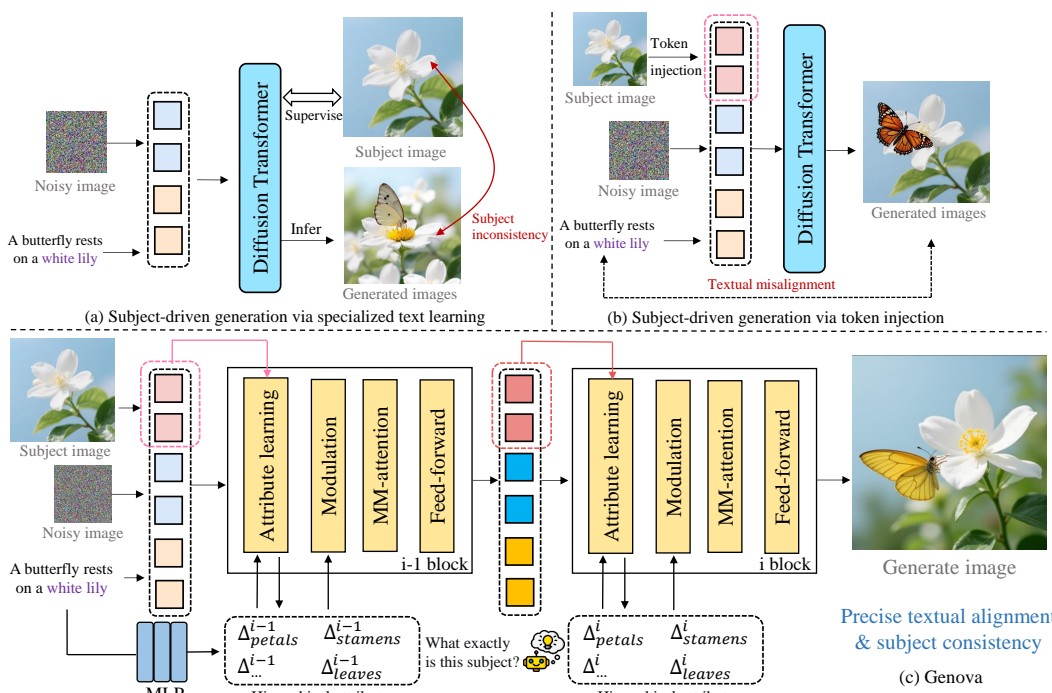

Figure 1: Comparison of our method **Genova** with two types of existing subject-driven generation methods. (a) The first-type methods achieve subject-driven generation via specialized text embeddings. These methods struggle with maintaining subject consistency. (b) The second-type methods achieve subject-driven generation via token injection. These methods heavily depend on the subject image and face challenges in text alignment. (c) In contrast, our method achieves both subject consistency and text alignment through hierarchical attribute learning for enhanced modulation.

preserve texture details, typically through model fine-tuning or introducing LoRA (Hu et al., 2022) (Low-Rank Adaptation) on subject-paired data for adapting to new inputs. However, these methods result in a strong reliance on subject image tokens, achieving high fidelity but significantly degrading the generation quality of the base model and hindering effective textual alignment. An example is shown in Fig. 1 (b), the lily flower is a simple "copy-paste", and its interaction with the butterfly looks unnatural.

To tackle the aforementioned dual challenge, we propose a personalized image generation framework called **Genova**. The core of Genova utilizes hierarchical subject tokens to fully leverage the base model's subject-text alignment capabilities for precise personalization. In DiTs, text-stream modulation (Peebles & Xie, 2023) is an effective way to align image generation with the conditioning inputs and avoid disrupting the sampling trajectory. We propose an innovative attribute learning module and insert it before modulation in each DiT block. Our proposed attribute learning module considers leveraging subject image tokens at multi-layer DiT blocks, which encode hierarchical features of the reference image. Specifically, it utilizes the output subject tokens from the previous block to improve text modulation via subject-driven self-attention. Due to the DiT model being trained on massive data, these tokens inherently possess certain text-alignment capabilities, and thereby the proposed attribute learning significantly enhances the semantic understanding of the subject in the text stream. This alleviates the issue shown in Fig. 1(a) where the personalized word embeddings poorly preserve object details. Furthermore, to reduce the over-reliance on the subject image token, which is shown in Fig. 1(b), we propose a Dropout strategy in which the subject image token is probabilistically excluded from the attention computation in the original DiT block during training.

Additionally, we identify two critical limitations in current data construction methods for subject-driven image generation (Tan et al., 2025; Guo et al., 2025; Wu et al., 2025): (1) insufficient fine-grained subject descriptions and (2) a lack of the modeling of object relationships. To address

these issues, we introduce a synthetic subject-paired dataset CoupleX that provides richer subject descriptions and emphasizes interactions within natural scenes, thereby improving the semantic understanding in generated images.

Our main contributions are summarized as follows:

- We propose **Genova**, a novel subject-driven image generation framework that incorporates a proposed attribute learning module. Through enhancing the semantic understanding of the subject, Genova achieves subject-text consistent personalized image generation.

- We present a synthetic dataset **CoupleX** to address the gap in existing datasets, which lack fine-grained descriptions and object interactions, thereby supporting a more accurate subject-semantic understanding in model training.

- Experimental results demonstrate that our method achieves both **textual alignment** and **subject consistency** in subject-driven generation and outperforms the state-of-the-art methods.

## 2 METHOD

In the following sections, we first present the preliminaries of the DiT model and modulation, then describe the design of our personalized image generation framework, **Genova**. Next, we present a specialized dropout strategy for DiT tuning, followed by an introduction to the new subject-paired dataset **CoupleX**.

### 2.1 PRELIMINARIES: ADDITIVITY PROPERTY AND MODULATION MECHANISM IN DITS

ToMe (Hu et al., 2024) demonstrates the semantic additivity of text embeddings in textual embeddings. By performing element-wise addition on the text "flower" for image generation, we can attribute it with additional characteristics, such as "white" + "flower" to generate a white flower. This finding can be extended to the modulation structure in modern text-to-image DiTs. Modulation refers to modifying the activations of a neural network on a per-channel basis, where each channel is multiplied by a single scale factor and shifted by a single bias scalar. As shown in Eq. 1, DiTs input the diffusion timestep $t$ and a pooled embedding of the text prompt $p$ into an MLP network to predict the aforementioned scale factor and bias scalar, collectively referred to as the $y$ vector:

$$y = \text{MLP}(t, \text{CLIP}(p)). \tag{1}$$

Specifically, instead of using the same modulation vector $y$ for all tokens, the text tokens associated with the target subject are modulated using specific modulation offsets that include its attribute:

$$y' = y + w\Delta_{attribute}. \tag{2}$$

Here, $w$ is a scale factor. Modulation mechanism leverages the additivity property to enable a precise integration of visual attribute representation into specific words without interfering with the denoising process.

### 2.2 HIERARCHICAL TEXT MODULATION ENHANCEMENT

As illustrated in Fig. 2, the proposed personalized image generation framework **Genova** is based on the FLUX (Labs, 2024) with a proposed attribute learning module. To enable the model to obtain the precise semantic understanding of all the subject attributes, Genova utilizes the subject image tokens to enhance the text-stream modulation in each block. Since subject image tokens inherently possess text-image alignment capabilities, the insight of the attribute learning module lies in leveraging hierarchical features encoded in subject image tokens within multi-layer DiT blocks to learn the attribute offsets of specific objects. First, Genova converts the text input into the initialized attribute token features $f^t$ through CLIP (Radford et al., 2021) and a single MLP:

$$f^t = \text{MLP}(\text{CLIP}(p)). \tag{3}$$

Next, Genova predicts a personalized offset $\Delta_{attribute}$ in the modulation space through Eq. 2. For each block, we predict modulation offsets $\Delta^i_{attribute}$, forming the set $\{\Delta^i_{attribute}|i = 1, 2, \ldots, N\}$.

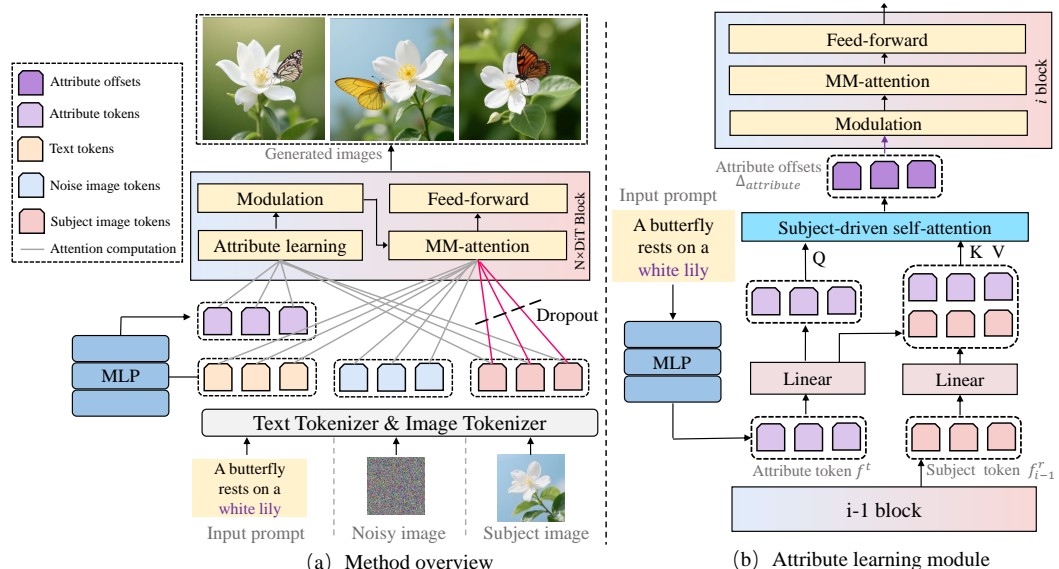

(a) Method overview  (b) Attribute learning module

Figure 2: (a) Overview of our proposed Genova framework. The text tokens, noisy image tokens, and subject image tokens are input into the DiT model. Each DiT block includes a proposed attribute learning module, followed by the modulation mechanism. Then the modulation offsets $\Delta_{attribute}$, which reveals that the specific subject attributes are applied to enhance the control of the semantic text in MM-attention. (b) Details of the attribute learning module. This module processes subject image tokens (from block $i$-1) and attribute tokens through subject-driven self-attention, which enhances the semantic understanding of the attribute tokens by incorporating hierarchical texture features from the subject image.

$N$ is the number of blocks in DiT. Such personalized modulation offsets are computed based on the given subject attribute embeddings and the subject image tokens in the previous layer:

$$\Delta^i_{attribute} = \mathcal{F}(f^t, f^r_{i-1}). \tag{4}$$

Here, $f^t$ is the embedded attribute tokens via the MLP network and $f^r_{i-1}$ represents the output token features of the reference subject image from the previous block. For the function $\mathcal{F}$, we use subject-driven self-attention to enhance attribute offsets $\Delta_{attribute}$ with the subject image tokens:

$$K = W^K[f^t \oplus f^r_{i-1}], V = W^V[f^t \oplus f^r_{i-1}],$$
$$Q = W^Q f^t, \ \mathcal{F}(f^t, f^r_{i-1}) = \text{softmax}(\frac{QK^T}{\sqrt{d}})V. \tag{5}$$

Here, $\oplus$ indicates matrix concatenation along the token axis; $d$ denotes the dimension; $W^Q$, $W^K$, $W^V$ represent the learnable linear transformation matrices. Finally, the hierarchical attribute offsets for text modulation can be formulated as:

$$y'_i = y + w \cdot \mathcal{F}(f^t, f^r_{i-1}). \tag{6}$$

Note that these operations are performed only on specific tokens in the prompt that are related to the subject. This structured composition of the subject-driven text offset facilitates more precise and adaptive control over the influence of text conditioning during the generation process. An example is shown in Fig. 2, Genova predicts $\Delta_{attribute}$ of "white petals", "sparse stamens" and "green leaves" for the "white lily" subject from the given image.

## 2.3 TUNING DITS WITH VERSATILE CONTROLS

Prior studies (Wu et al., 2025; Mou et al., 2025; Tan et al., 2025) on subject-driven generation via token injection encounter a challenge of "copy-paste" while training on the subject-pair data. This can be attributed to the inherent characteristics of gradient backpropagation, which predisposes neural networks to over-rely on specific subject tokens rather than acquiring their semantic representations.

A direct solution for such "identity mapping" is to add perturbations to the subject images before reconstructing the subjects, including combining two subjects to build a complex paired dataset (Wu et al., 2025; Guo et al., 2025) and making data augmentations. However, the hand-crafted perturbation strategies may not be universal across domains and datasets. In our Genova framework, we turn to utilize the simple and elegant Dropout techniques (Srivastava et al., 2014). Specifically, we employ Dropout to discard the activation of subject tokens with a probability of $p$. The Dropout mechaism is applied in per-block for each sample. Without introducing any additional modules or parameters, this simple strategy facilitates effective training of our module, thereby enhancing DiT's ability to learn semantic representations of the subject image, which in turn enables the learned subject to interact with words in the text latent space.

Our method uses UNO (Wu et al., 2025) as the base DiT model. In our training, we fix the parameters of the base model and only train the parameters of the MLP layer in Eq. 3 and those in the attribute learning module. With a fixed probability of $p = 0.2$, the subject image tokens are skipped in the DiT block's attention computation and are only used in the attribute learning module.

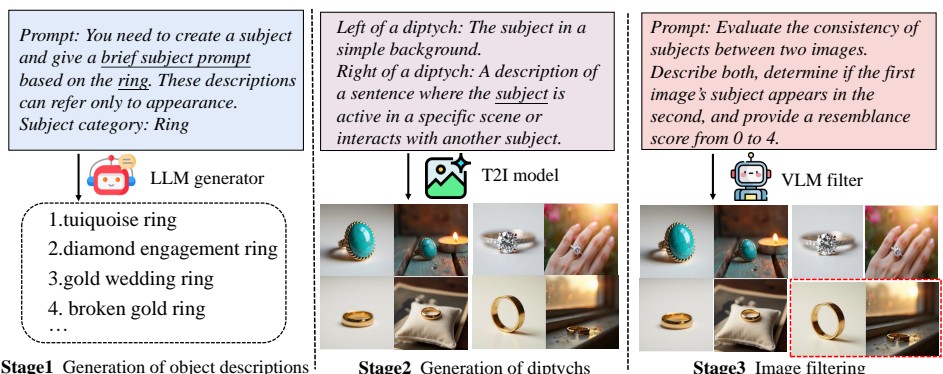

Figure 3: Illustration of the data construction pipeline. The red dashed boxes indicate the filtered samples.

### 2.4 TRAINING DATA SYNTHESIS

To further enhance the model's text semantic understanding capability, we propose a synthetic subject-pair dataset CoupleX and the pipeline for data construction is illustrated in Fig. 3. Our data construction pipeline consists of three stages. First, we leverage a large language model (LLM) to generate diverse subject appearances based on specified category descriptions. Next, we use a vision language model (VLM) that produces a diptych prompt for each subject: (1) a reference image depicting the subject against a simple background, and (2) a target image showing the subject engaged in activities or interactions with other subjects. The prompt is then input into the text-to-image (T2I) model to obtain the paired images. Finally, a VLM-based filtering process is applied to automatically remove low-quality data, such as the case of the red dashed box in Fig. 3 where the golden ring on the right is broken while and the left one is intact. Unlike existing datasets (Wu et al., 2025; Tan et al., 2025; Guo et al., 2025) that primarily focus on the object similarity in diptychs, our method emphasizes (1) detailed descriptions of target subjects, (2) their interactions with surrounding objects and activity representation in complex scenes. The objects in CoupleX dataset include 295 categories from Object365 Shao et al. (2019) (with uncommon categories removed), comprising a total of 45,548 paired data. The detail progress and data samples can be seen in the supplementary materials.

## 3 EXPERIMENTS

### 3.1 EXPERIMENTS SETUPS

**Implementation Details.** We adopt UNO (Wu et al., 2025) as our base model for subject-driven image generation. The DiT backbone of UNO contains $N = 57$ blocks. The proposed attribute module is inserted into all double-stream and single-stream blocks. We train the parameters of MLP layers and the parameters of subject-driven self-attention with an AdamW optimizer. We set the learning rate at $1e^{-5}$ and conduct the training on 4 A800 GPUs.

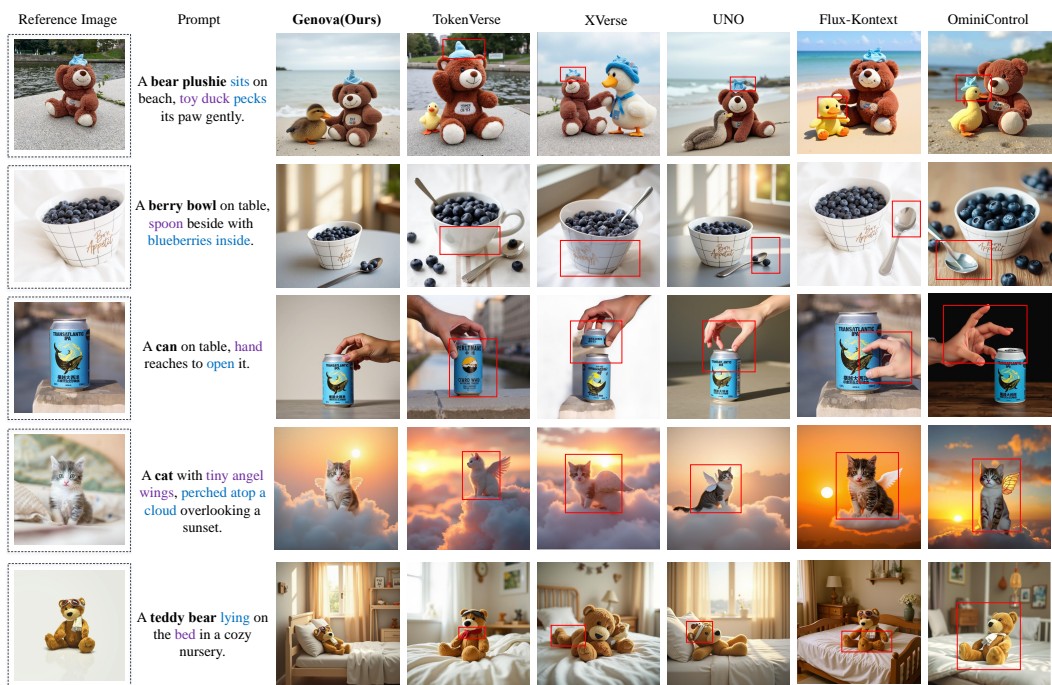

Figure 4: **Qualitative comparison.** In the prompt, the bolded text represents the subject, the purple text indicates another object with which the target subject interacts, and the blue text represents the action. The red bboxes in the generated images highlight the misalignment of the reference image or the prompt. Other methods face a trade-off between textual alignment and subject consistency, whereas our proposed Genova achieves both controllable and flexible results simultaneously.

**Comparison Methods.** We compare our method with state-of-the-art personalization approaches for subject-driven generation, including two categories: (1) subject-driven generation methods based on subject token injection, represented by OminiControl (Tan et al., 2025), UNO (Wu et al., 2025), Flux-Kontext (Labs et al., 2025b), and (2) subject-driven generation methods based on specialized text embedding learning, represented by XVerse (Chen et al., 2025a) and TokenVerse (Garibi et al., 2025).

**Evaluation Benchmarks.** Current benchmarks fall short in evaluating the deep semantic understanding of target subjects. To address this gap, we construct a custom dataset from Dream-Bench (Ruiz et al., 2023a), DreamBench++ (Peng et al., 2024), and online sources that encompass 50 subject images. For prompt generation, we utilize a LLM to depict the interactions and activities of the target subject with other entities.

**Metrics and User Study.** We follow prior methods (Wang et al., 2024; Huang et al., 2025; Sun et al., 2024; Garibi et al., 2025) to evaluate our personalization method from two perspectives: the fidelity of the generated concepts (Concept Preservation, noted as CP) and the alignment between the generated images and the textual prompt (Prompt Fidelity, noted as PF). Since there is a certain trade-off between these two metrics, their product CP·PF is also included following previous work (Peng et al., 2024) since it better reflects the model's comprehensive understanding of the subject image. The metrics CP, PF and CP·PF consist of two scores, including evaluations from both the VLM (CP-L, PF-L, CP·PF-L) and human reviewers (CP-H, PF-H, CP·PF-H). The human evaluation includes 45 volunteers who are asked to score the generated images in terms of concept preservation and prompt fidelity. To further evaluate image-text alignment, we measure the similarity between their CLIP embeddings (CLIP-T).

## 3.2 COMPARISONS

**Qualitative Comparison.** The qualitative comparison of single-subject condition generation results is shown in Fig. 4. As observed, the first-type methods, represented by UNO, OminiControl, and Flux-Kontext, preserve object details well but fail to fully capture the content specified by the

Table 1: **Quantitative comparison**. CP-L and PF-L scores evaluate concept preservation and image-text alignment by VLM, respectively. CP-H and PF-H scores evaluate concept preservation and image-text alignment by humans, respectively. CLIP-T also evaluates image-text alignment. All scores range from 0 to 1. ↑: higher is better.

| Method | CP-L↑ | PF-L↑ | CP·PF-L↑ | CLIP-T↑ | CP-H↑ | PF-H↑ | CP·PF-H↑ |
|---|---|---|---|---|---|---|---|
| Ominicontrol (Tan et al., 2025) | 0.695 | 0.755 | 0.525 | 0.2743 | 0.666 | 0.748 | 0.499 |
| UNO (Wu et al., 2025) | 0.803 | 0.747 | 0.600 | 0.2674 | 0.698 | 0.645 | 0.451 |
| Flux-Kontext (Labs et al., 2025a) | **0.866** | 0.728 | 0.630 | 0.2717 | **0.881** | 0.749 | 0.659 |
| TokenVerse (Garibi et al., 2025) | 0.646 | 0.792 | 0.512 | 0.2557 | 0.543 | 0.607 | 0.330 |
| XVerse (Chen et al., 2025a) | 0.734 | 0.743 | 0.545 | 0.2738 | 0.606 | 0.629 | 0.381 |
| **Genova (Ours)** | 0.785 | **0.843** | **0.662** | **0.2757** | 0.847 | **0.847** | **0.717** |

prompt. For example, in row 1, UNO and OminiControl show the toy duck pecks the "face" rather than the "paw" of the bear plushie, while both Flux-Kontext and OminiControl in row 2 fail to depict the "spoon beside blueberries". In row 3, none of these methods successfully reflect the prompt "open a can". In contrast, the second-type methods, represented by TokenVerse and XVerse, focus on specialized text embedding learning. They generate images that generally align with the prompt but poorly maintain object details. For instance, the shape of the doll's hat in row 1, the color of the cat in row 4, and the appearance of the vase and flowers in row 5 are all inaccurately maintained. These observations indicate that these methods struggle to capture the finer attributes of objects. In comparison, our method achieves both high fidelity and textual alignment, demonstrating its effectiveness in capturing the deep semantic understanding of the subject.

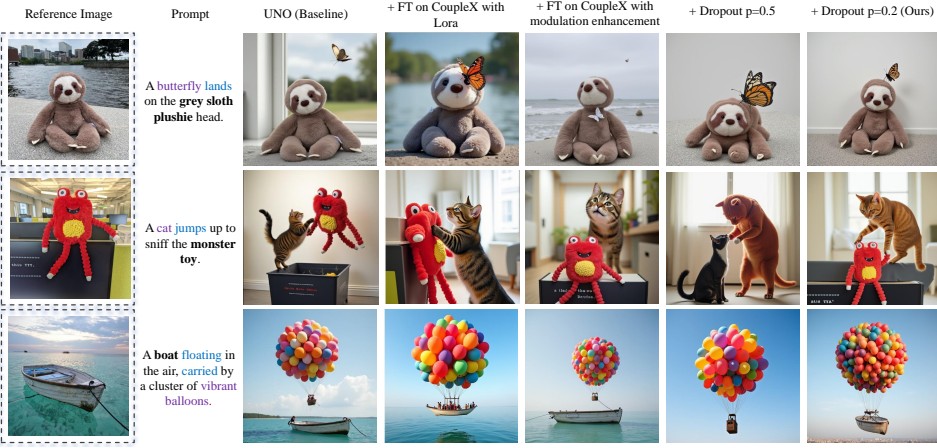

Figure 5: **Ablation study.** Fine-tuning the baseline method on our proposed dataset improves the model's understanding of object interactions. The introduction of modulation enhancement with an appropriate Dropout strategy ($p$=0.2) significantly improves subject and textual consistency in generated images.

**Quantitative Comparison.** We present the quantitative comparison results in Tab. 1. The first-type methods generally exhibit strong conception preservation, with Flux-Kontest obtaining the highest CP-L score of 0.866 and CP-H score of 0.881. In contrast, although second-type methods like TokenVerse and XVerse score higher on prompt fidelity, they show weaker conception preservation, resulting in lower CP·PF scores. Our method strikes a more effective balance, achieving the highest CP·PF-L score of 0.662 and CP·PF-H score of 0.717. Additionally, CLIP-T is insensitive to variations in image-text alignment across all methods, with scores around 0.27; our method leads slightly with a score of 0.2757. These results demonstrate that our proposed Genova achieves subject-text consistent personalized image generation.

### 3.3 EXTENSION TO MULTIPLE-SUBJECT DRIVEN GENERATION

Despite only conducting training on single-subject pair datasets, our method demonstrates a promising adaptability to multi-subject driven image generation. Utilizing UNO (Wu et al., 2025) as the

base model which supports input of multiple subjects, we interject our proposed attribute learning module into its DiT architecture. As illustrated in Fig. 11, our method significantly enhances the model's semantic understanding of objects, facilitating the subject consistency and natural interactions with other entities. For example, row 1 in Fig. 11 shows UNO fails to represent the action of "insert" while our method integrates these two subjects naturally. These outcomes underscore the extensibility and generalizability of our method.

Table 2: **Ablation study**. The upper group demonstrates the effectiveness of the proposed CoupleX dataset, while the lower group showcases the effectiveness of the proposed Genova framework. CP and PF scores evaluate concept preservation and image-text alignment by VLM, respectively. CLIP-T evaluates image-text alignment. All scores range from 0 to 1. ↑: higher is better.

| Method | CP↑ | PF↑ | CP·PF↑ | CLIP-T↑ |
|---|---|---|---|---|
| Baseline | **0.803** | 0.747 | 0.600 | 0.2674 |
| + FT on CoupleX with Lora | 0.734 | 0.843 | 0.619 | **0.2787** |
| Baseline | **0.803** | 0.747 | 0.600 | 0.2674 |
| + FT on CoupleX with modulation enhancement | 0.667 | 0.700 | 0.467 | 0.2669 |
| + Dropout $p$=0.5 | 0.790 | 0.748 | 0.591 | 0.2763 |
| + Dropout $p$=0.2 (Ours) | 0.785 | **0.843** | **0.662** | 0.2757 |

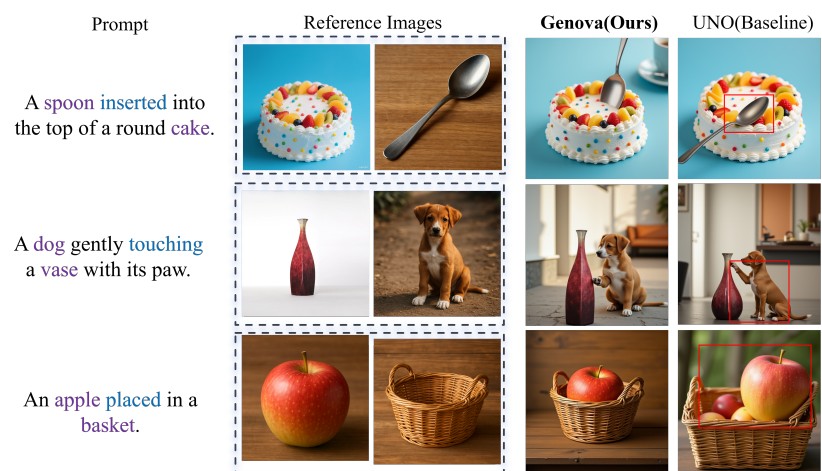

Figure 6: Our method can be extended to **multi-subject driven generation** and show better semantic understanding of the subject interaction than the baseline.

## 3.4 ABLATION STUDY

We conduct an ablation study to evaluate the effectiveness of our proposed modules and use UNO (Wu et al., 2025) with its public checkpoint as the baseline method. Quantitative results and qualitative results are shown in Tab. 2 and Fig. 5, respectively. Ablation studies are divided into two groups: the first group (first two rows in Tab. 2) is used to demonstrate the effectiveness of the proposed CoupleX dataset, and the second group is used to prove the effectiveness of our designed Genova. Comparing rows 1 and 2, applying LoRA fine-tuning (FT) on UNO using CoupleX dataset boosts the PF metric from 0.747 to 0.843, demonstrating that the proposed dataset enhances model comprehension of the subject. Row 4 in Tab. 2 indicates adding the proposed attribute learning modules to enhance the text-stream modulation in DiT blocks. In contrast to the setting in row 2, the parameters in the proposed module are initialized while keeping the pre-trained model fixed. Due to over-reliance on the subject image tokens, the model fails to adequately learn these initialized modules, resulting in performance degradation. To address this, the introduction of an appropriate Dropout rate ($p$=0.2) on the subject image tokens unlocks the full potential of the modulation enhancement, significantly improving subject and textual consistency in generated images, with the highest CP·PF of 0.662. From column 5 in Fig. 5, an excessively high Dropout rate ($p$=0.5) leads to a loss of textural details.

## 4 RELATED WORK

### 4.1 SUBJECT-DRIVEN GENERATION VIA SPECIALIZED TEXT LEARNING

In the subject-driven generation task, one approach extends the pre-trained model's distribution to incorporate specific subjects or concepts from the given image sets by learning pseudo-words or specialized text embeddings. Pioneered by DreamBooth (Ruiz et al., 2023b) and Textual Inversion (Gal et al., 2022), this paradigm enables personalized text-to-image generation through pseudo-word embeddings learning. Disenbooth (Chen et al., 2023) addresses the impact of irrelevant factors such as background or pose on specialized text embeddings by designing weak denoising and contrastive embedding auxiliary tuning objectives for disentangling subject identity from irrelevant information. However, a key limitation of these methods is their requirement for per-image-set optimization, which presents non-trivial challenges: insufficient training yields low fidelity, while excessive tuning may degrade the base model. Recently, TokenVerse (Garibi et al., 2025) proposes a disentangled multi-concept personalization method that learns offsets between target objects and their global class in text latent space, rather than pseudo-words from scratch. Then the offsets are used for per-token modulation in the DiT model. However, TokenVerse exhibits instability due to the per-token optimization process. XVerse (Chen et al., 2025a) and ModAdapter (Zhong et al., 2025) use a tuning-free approach, leveraging CLIP (Radford et al., 2021) to extract high-level semantic image features and employing a cross-attention mechanism to enhance text-stream modulation. However, their reliance on CLIP for feature extraction, which lacks fine-grained discriminability, causes difficulties in precisely capturing detailed object attributes and leads to biases in the semantic interpretation of subjects.

### 4.2 SUBJECT-DRIVEN GENERATION VIA TOKEN INJECTION

The other approach to achieving subject-driven generation is to inject tokens of the given subject into the pre-trained text-to-image model. Early works trained an encoder to extract features from the subject image and fed them into the pre-trained U-Net-based model (Rombach et al., 2022). BLIP-Diffusion (Li et al., 2023) utilizes BLIP-2 to extract image features and employs the Q-Former to efficiently align image and text features. IP-Adapter (Ye et al., 2023) designs a decoupled cross-attention module that injects image features as an independent condition into the U-Net's attention layers to maintain textual control. InstantBooth (Shi et al., 2024) introduces adapter layers to learn visual features and injects them into the model's intermediate layers while keeping the base model fixed. Recently, in the field of image generation, the performance of the DiT model (Labs, 2024; Labs et al., 2025a) has surpassed that of U-Net-based models (Rombach et al., 2022). Omini-Control (Tan et al., 2025) proposes a unified sequence processing strategy that combines condition tokens with subject image tokens for token interactions. Building on this, UniReal (Chen et al., 2025b) introduces an image index embedding for different categories of input images, effectively distinguishing different subjects. Moreover, DreamO (Mou et al., 2025) extends the strategy of unified sequence processing to multiple downstream tasks, including identity-driven generation and try-on image customization. These works generally use LoRA to fine-tune the pre-trained DiT model, eliminating the need for tuning during inference. Despite these advancements, the fine-tuned attention mechanism may overly rely on the injected subject token, thereby compromising the base model's text alignment capability. In this study, we consider using hierarchical subject image tokens to assist in precise semantic understanding, achieving high-fidelity generation while fully leveraging the text alignment advantages of the base model.

## 5 CONCLUSION

In this paper, we analyzed two types of subject-driven generation methods and their trade-off on the textual alignment and the consistency of the given subject. To address this, we introduced the framework Genova enhanced with a new attribute learning module. This module utilizes hierarchical subject image tokens to enhance the learning of subject attributes and aggregate them into the precise textual inversion, thereby significantly improving the model's ability to semantically comprehend subject images. This enables our method to achieve controllable and flexible subject-driven image generation. Additionally, we introduced a synthetic dataset CoupleX designed to focus on interactions among subjects, which helps in enriching the contextual understanding of generated images. In future work, we will expand our dataset to include multi-subject paired data and train Genova on it to enhance its application potential.

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

APPENDIX

In the appendix, we first introduce the details of the user study. Then, we detail the construction pipeline of the proposed CoupleX dataset. Next, we discuss the limitations. Finally, we present more qualitative results from both single-subject personalization and multiple-subject personalization.

## A    USER STUDY DETAILS

Fig. 7 shows the user study interface for evaluating conception personalization (CP-H) and prompt fidelity (PF-H). Each case includes two questions, one on conception personalization and the other on prompt fidelity. Participants rate their responses on a 5-point scale: (1) "Very inconsistent", (2) "Somewhat inconsistent", (3) "Fair", (4) "Quite consistent", and (5) "Very consistent".

Figure 7: Screenshot of our user study rating interface.

## B    DATASET CONSTRUCTION PIPELINE

In this section, we introduce our dataset construction pipeline with the Large Language Model (LLM) and Vision Language Model (VLM). For the LLM, we use Qwen3-14B, and for the VLM, we use Qwen2.5-VL-32B-InstructIn. In Fig. 8, we present the prompt for data construction.

In Stage 1, LLM generates 50 brief subject prompts based on given subject categories. Each prompt must be realistic, concise, and created using common sense, with no repetition of subjects. The category of the subject is uniformly sampled from the 295 categories selected from the Object365 (Shao et al., 2019) taxonomy tree. In Stage 2, a short and vivid sentence is written using the given subject by LLM. This sentence should describe a plausible scene or interaction involving the subject, maintaining the exact wording of the original subject without omission or reordering. Then, we use Flux model to generate images based on the sentence descriptions above. In Stage 3, VLM evaluates the consistency of the subject between two generated images. It assigns a resemblance score from 0 to 9 based on detailed visual comparison, where a score below 2 results in the image being filtered out. To further improve the data quality, we apply manual filtering after VLM filtering. From Fig. 9(b), we generated 72,000 paired images, of which 50,996 remained after filtering with the VLM filter.

Due to the efficiency of the data synthesis pipeline, the manually filtered samples account for approximately 10%. In the end, the proposed CoupleX dataset consists of 45,548 high-quality paired images. CoupleX contains 295 categories, which can be grouped into 11 parent categories. The number of images in each parent category is shown in Fig. 9(a).

In Fig. 10, we showcase four representative examples from our proposed CoupleX dataset. CoupleX is enriched with diverse subjects, each accompanied by specific descriptions. For instance, it includes items such as "cooked shrimp" and "sliced hamimelon". Additionally, our dataset includes common activities of subjects, such as "jellyfish drifting" and "zebras sipping water". These features are absent in previous synthetic subject-pair datasets. This enhancement in both the variety of subjects and the incorporation of subject behaviors enables a more detailed and applicable range of scenarios for personalization image generation.

---

### Dataset Construction Pipeline

***Stage 1: Generation of suject descriptions(LLM)***
Role: Please be very careful and generate 50 breif subject prompts for text-to-image generation.
You will be given a [subject category], based on which you are required to create a brief subject prompt that describes a plausible, real-world entity. The description should focus solely on visual characteristics, and must be grounded in common sense. Repetition across subjects is not allowed.
Example [subject category]: Cat, [subject1]: British Shorthair cat [subject2]: A yellow furry cat...

***Stage 2: Generation of diptychs(LLM)***
Role: You are an AI expert. Please generate a vivid, concise sentence for text-to-image generation.
Requirements:
1. Use the given subject.
2. Description of a sentence about a subject that can be active in a certain scene or interact with another object.
3. Include specific background or setting details, but not too complex.
4. The sentence must include all words in the subject exactly as provided, with no omissions or reordering.

***Stage 3: Image Filtering(VLM)***
Role: You are an AI expert tasked with objectively assessing the consistency of subjects across two images. You will analyze two images. Describe each image and determine if the subject from the first is present in the second.
Step 1: Thoroughly inspect the most prominent subject in both images. Deconstruct it into key evaluative components; however, you do not need to include these in your output.
Step 2: Conduct a detailed comparison of each identified component, noting all differences. These details do not need to be listed in your output.
Step 3: Based on your comprehensive analysis, provide a single integer score from 0 to 9 that reflects the overall similarity of the subject between the two images.
Output Format: Only output a single integer from 0 to 9, and nothing else.
**The image with a score less than 2 will be filtered out.**

---

Figure 8: The dataset construction pipeline with LLM/VLM.

## C    LIMITATION AND DISSCUSION

Due to the limitations of the foundation model, the generated paired images cannot fully replicate all real-world scenarios. This constraint subsequently restricts the associative abilities of the proposed Genova. Moreover, our method currently does not extend to tasks involving the generation of three or more objects interacting with each other, as there is a lack of high-quality datasets featuring

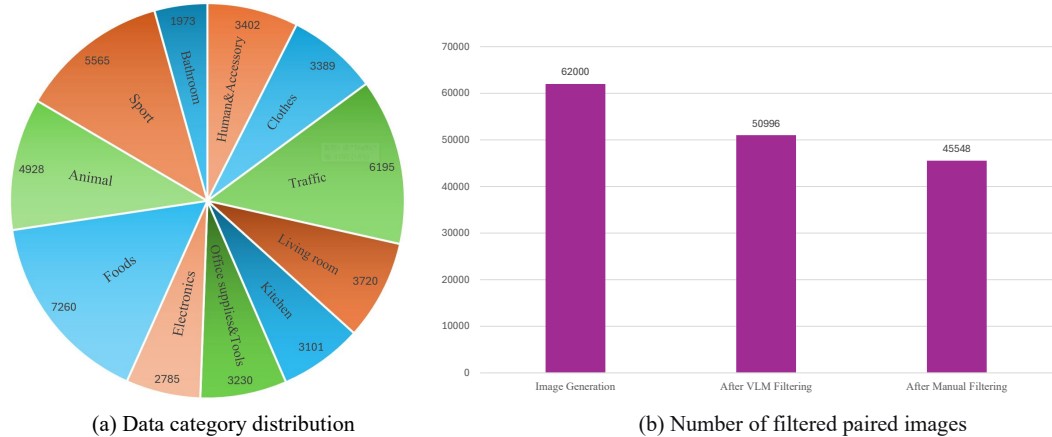

(a) Data category distribution       (b) Number of filtered paired images

Figure 9: (a) Data category distribution and (b) the number of generated images after VLM and manual filtering steps.

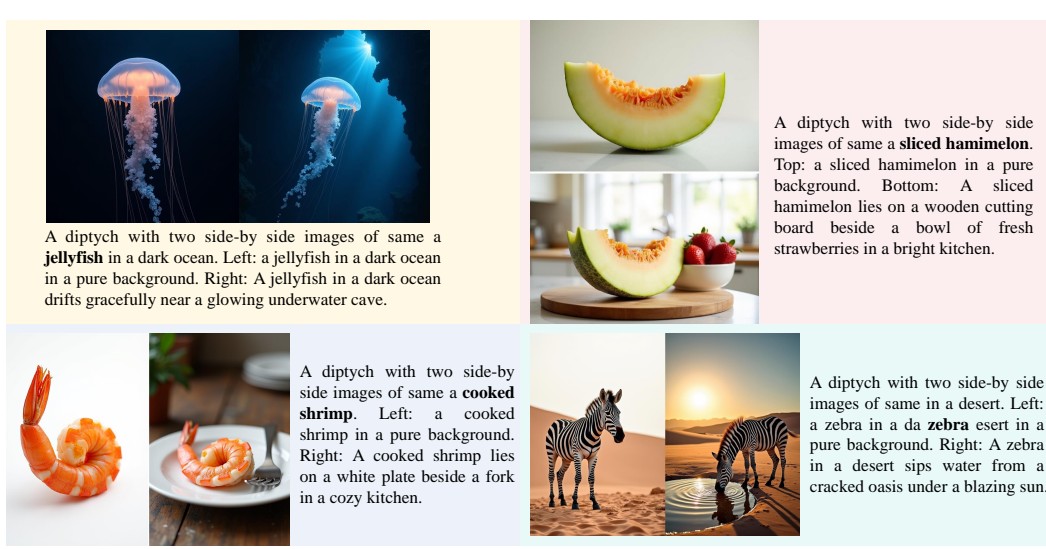

Figure 10: Examples from our proposed CoupleX dataset. The bold texted text represents the subject.

multiple interacting objects. In future work, we will expand our dataset to include multi-subject paired data and train Genova on it to enhance its application potential for personalization on more subjects.

## D  MORE QUALITATIVE RESULTS

We show a more qualitative comparison of single-subject personalization in Fig. 12. The baseline methods encounter significant challenges with subject consistency and prompt alignment. For instance, in row 4 of Fig. 12, although TokenVerse successfully places a rose into a vase, the shape of the vase is changed. UNO and Flux-Kontext maintain the target vase better, but the generated image merges the rose and the vase into an unnatural composite. Besides, XVerse and OminiControl exhibite obvious breakdowns in their output. Similarly, in row 3, while TokenVerse, XVerse, and UNO do generate images of a "foggy park", the shape of the street lamp is noticeably altered. Neither Flux-Kontext nor OminiControl accurately reflects the "foggy" aspect specified in the prompt. In

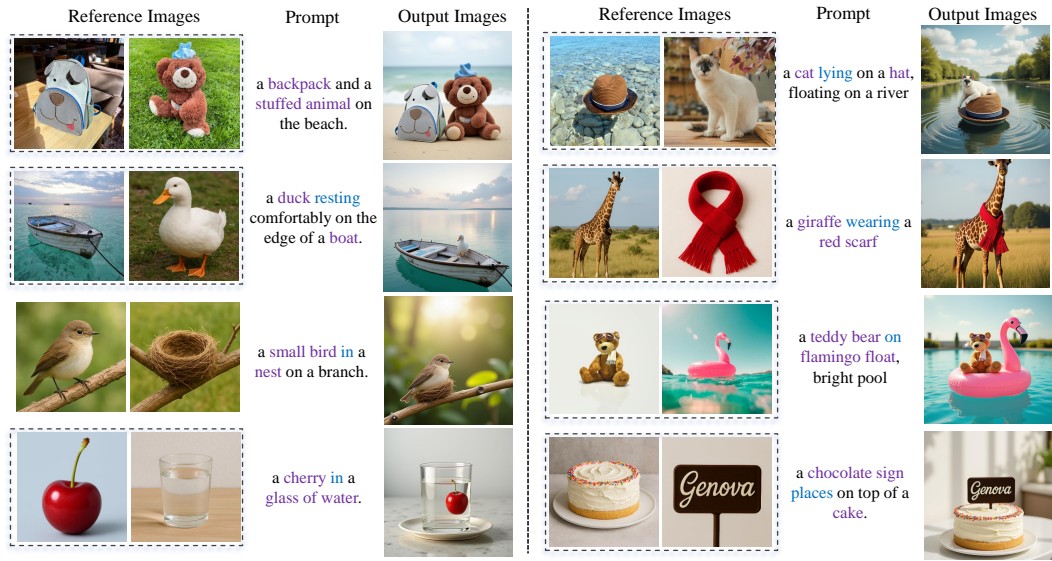

Figure 11: More results of multi-subject image generation. In the prompt, the purple text indicates the subjects, and the blue text represents the interaction.

contrast, our method achieved outstanding performance in maintaining high fidelity and semantic consistency, demonstrating its superiority against the challenges faced by the baseline models.

For multi-subject image generation, we present additional results generated by Genova in Fig. 11. These examples demonstrate Genova's capability to preserve detailed features of subjects, such as the teddy bear's goggles in row 3 and the "Genova" chocolate brand in row 4. Furthermore, it adeptly combines the given two objects according to the prompt, as seen with the giraffe wearing a red scarf in row 2, and the cherries submerged in a water glass in row 4. These results highlight the model's proficiency in maintaining subject consistency while creatively integrating distinct subjects as specified in the prompts.

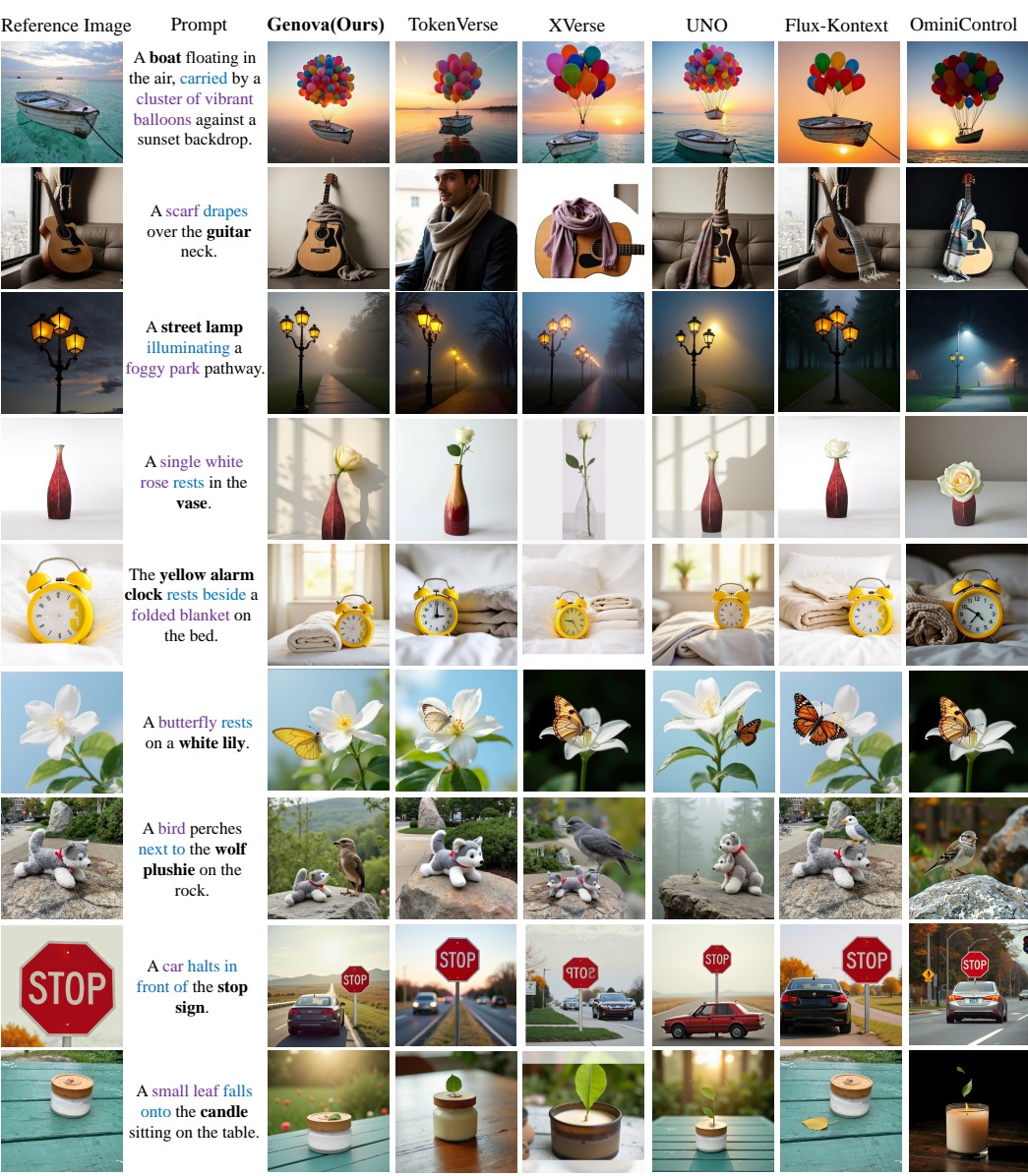

Figure 12: More qualitative comparison of subject-driven image generation. In the prompt, the bolded text represents the subject, the purple text indicates another object with which the target subject interacts, and the blue text represents the action.

