# OpenReview forum: "Towards Subject-Consistent and Text-Aligned Personalized Image Generation via Precise Attribute Learning"
_ICLR.cc/2026/Conference — Submitted to ICLR 2026_

### Official Review · Reviewer_MZfB · 2025-10-25

**Soundness:** 3
**Presentation:** 3
**Contribution:** 2
**Rating:** 4
**Confidence:** 3

**Summary:**

This paper aims to achieve a balance between text prompts and original image features in
personalized image generation. Prior works tend to overemphasize either the text prompt—resulting in
loss of fine image details—or the original image—leading to stiff, less semantically aligned results. To
address this, the authors propose Genova, a framework designed to ensure both text alignment (Text-
Aligned) and subject consistency (Subject-Consistent) in generated images.

**Strengths:**

1. The paper clearly identifies a key challenge in subject-driven generation: the trade-off between
textual alignment and subject consistency.
2. The proposed attribute learning module is novel in its attempt to integrate hierarchical subject
features into the modulation process.
3. The introduction of the CoupleX dataset addresses a meaningful gap in existing data, particularly
in capturing fine-grained interactions.
4. The use of dropout as a regularizer is simple yet effective, as shown in the ablation study.

**Weaknesses:**

1. Unclear Contribution of the Core Module:
The ablation study (Table 2) reveals a critical issue: adding the proposed attribute learning module
alone leads to a significant performance drop (CP-PF from 0.600 to 0.467). Performance is
recovered and surpasses the baseline only when combined with dropout. This strongly suggests
that the observed improvements may be primarily attributable to the regularization effect of
dropout rather than the intrinsic design of the module itself. The authors should provide further
evidence to disentangle these contributions, such as a sensitivity analysis across dropout rates or
experiments with alternative regularization methods.
2. Questionable Module Design:
The design of the subject-driven self-attention, where K and V are derived from the concatenation
of attribute and subject tokens while Q comes only from attribute tokens, deviates from standard
practice. This design risks creating overly strong correlations between the input and output,
potentially leading to rank collapse in the attention matrix and encouraging the model to learn a
simplistic identity mapping rather than a semantically meaningful alignment. The performance
drop without dropout supports this concern.
3. Limited and Subjective Evaluation Benchmark:
The reliance on a custom benchmark and metrics (CP-L/PF-L, CP-H/PF-H) that are heavily based
on VLM and human evaluation makes it difficult to objectively compare the method against the
broader literature. The inclusion of more established, objective metrics (e.g., CLIP-I, DINO-Vis) is
necessary for better reproducibility and fair comparison. The CLIP-T scores are also too close
across all methods to be discriminative.

**Questions:**

1. Could the authors provide further justification for the specific Q/K/V design in the attribute learning
module? Were alternative designs (e.g., using subject tokens only for K/V) explored, and if so,
what were the results?
2. Given that the module alone degrades performance, what specific functionality does it provide
that, when regularized by dropout, leads to improvement? Can the authors design an experiment
to isolate the module's contribution from that of the dropout?
3. Would the authors consider adding more standard evaluation metrics (e.g., CLIP-I, DINO) to
facilitate a more direct comparison with future work?

---

### Official Review · Reviewer_2kwM · 2025-10-27

**Soundness:** 4
**Presentation:** 3
**Contribution:** 2
**Rating:** 4
**Confidence:** 5

**Summary:**

This work addresses the challenge in the field of personalized image generation, where it is difficult to simultaneously ensure both textual alignment and the fidelity of reference subjects. To overcome this, the authors introduce an innovative attribute learning module. Additionally, they construct the synthetic dataset CoupleX, which features subject-paired samples focused on depicting activities and interactions within natural scenes. The proposed method demonstrates promising results.

**Strengths:**

- The paper successfully identifies the limitations of the two previously proposed approaches: specialized text learning and token injection. It provides clear reasoning for the constraints observed in each method’s results.
- The paper is well-written and easy to follow.
-  A comprehensive introduction to the new subject-paired dataset, CoupleX.

**Weaknesses:**

- What is the significance of this work in the context of powerful base models, such as nano banana?

- From a macro perspective, the proposed framework is simply composed of specialized text learning and token injection. Consequently, the methodological novelty appears limited, and the contribution seems incremental.

- To validate the necessity of the attribute learning module, an ablation study should be conducted. This would involve removing the attribute learning module while keeping the design of the other components intact, and then using simple textual inversion to inject features for comparison

**Questions:**

In line 157, the 'text input' refers to the phrase 'A butterfly rests on a white lily' or the term 'white lily' itself.

---

> ### Comment · Reviewer_2kwM · 2025-11-24
>
> A considerable amount of time has elapsed, yet the authors still have not put forward any rebuttal, and thus I stand by my score.

---

### Official Review · Reviewer_Q5N3 · 2025-11-01

**Soundness:** 2
**Presentation:** 2
**Contribution:** 2
**Rating:** 4
**Confidence:** 4

**Summary:**

This paper introduces Genova, a DiT-based subject-driven generation framework featuring a novel attribute learning module. The paper claims that Genova can capture detailed subject attributes. Furthermore, the authors develop CoupleX, a synthetic dataset of subject-paired samples that focuses on activities and interactions within natural scenes. The results show that Genova achieves state-of-the-art (SOTA) performance in subject- and prompt-consistent personalized image generation.

**Strengths:**

The goal of capturing attribute-level information presents a strong motivation.

The hierarchical design is reasonable.

**Weaknesses:**

The main claimed contribution, attribute learning, is not adequately verified in this paper.

*1. Lack of Evidence for Attribute Learning:Why can the proposed Attribute Learning module actually learn attributes? I did not see any compelling evidence to support the claim that the module learns distinct attributes. For instance, the statement in Lines 208-210 ("shown in Fig. 2, Genova predicts $\Delta attribute$ of 'white petals,' 'sparse stamens,' and 'green leaves' for the 'white lily' subject from the given image") cannot be substantiated by an examination of Figure 2.

*2. Irrelevance of CoupleX Dataset:The proposed dataset, CoupleX, seems irrelevant to attribute learning. It appears to be a standard subject-driven generation dataset. Furthermore, the paper is missing a crucial comparison with other existing subject-driven datasets.3.

*3. Evaluation Benchmark:The evaluation benchmark lacks transparency. The paper relies on only one self-curated benchmark gathered from different sources, which contains only 50 subjects. The authors need to clarify: What is the total number of test cases? Why are results from other established benchmarks not reported?

**Questions:**

The main problem is that the paper does not sufficiently investigate or adequately support its core idea, which is: "Towards Subject-Consistent and Text-Aligned Personalized Image Generation via **Precise Attribute Learning**".

---

### Official Review · Reviewer_quGE · 2025-11-01

**Soundness:** 3
**Presentation:** 3
**Contribution:** 2
**Rating:** 6
**Confidence:** 5

**Summary:**

This paper proposes the Genova framework for personalized image generation. Specifically, to avoid the "copy-paste" issue, it uses dropout to weaken the direct injection of the reference image. Instead, it leverages image features to guide and enhance the modulation process of textual conditioning. This enables the model to more precisely understand the subject's visual attributes and combine them with the text description. Experiments demonstrate the effectiveness of the proposed method.

**Strengths:**

1. Instead of making incremental improvements on the two existing mainstream methods (text learning vs. token injection), it utilizes image features to enhance text modulation, cleverly bypassing the inherent drawbacks of both traditional approaches.

2. This paper points out the prevalent 'copy-paste' problem in 'token injection' methods and proposes using Dropout on the subject image tokens during training.

**Weaknesses:**

- The custom benchmark from DreamBooth may not contain challenging samples with precise attributes. Considering the large scale of the $45,548$ paired dataset, this raises concerns about simply overfitting to the custom benchmark using such a large dataset. Especially compared with ablation studies on dropout hyperparameters, the results with *0.2* largely outperformed those with *0.5*, indicating that the proposed method lacks effectiveness and robustness.

- Was there an experimental investigation into the dropout hyperparameter settings? The ablation study only shows the values 0.2 and 0.5.

- Only a few qualitative examples on multi-subject driven generation are provided. Regarding multi-concept results, a well-established benchmark already exists in [1].

- There is a citation format inconsistency on line 258.

$[1] \text{Kumari, Nupur, et al. "Multi-concept customization of text-to-image diffusion." Proceedings of the IEEE/CVF conference on computer vision and pattern recognition. 2023.}$

**Questions:**

- Regarding the motivation for learning precise attributes from hierarchical image tokens, why not directly use fine-grained attribute captions as baseline implementations for comparison?

---

### Meta-Review · Area_Chair_qvWm · 2026-01-06

**Summary:**

This work investigates personalized image generation by balancing textual alignment and reference subjects preservation. A DiT-based framework Genova is proposed to mitigate the challenge with an attribute learning module. Moreover, a dataset CoupleX featuring subject-paired samples is collect to fine-tune the backbone. Experiments shows that the proposed pipeline can trade between different objects effectively.

**Reviewer Concerns:**

The major concerns focused on the limited novelty compared to existing work, the performance drop from the proposed attribute learning module, insufficient evaluation benchmark and ablations for hyper-parameters. There is no rebuttal provided and those concerns have not been addressed.

**Reviewer Scores:**

Due to the lack of the rebuttal, the initial scores will not be changed.

---

### Decision · Program_Chairs · 2026-01-26

Reject